# Lung Recruitment Maneuvers Assessment by Bedside Lung Ultrasound in Pediatric Acute Respiratory Distress Syndrome

**DOI:** 10.3390/children9060789

**Published:** 2022-05-27

**Authors:** Mireia Mor Conejo, Carmina Guitart Pardellans, Elena Fresán Ruiz, Daniel Penela Sánchez, Francisco José Cambra Lasaosa, Iolanda Jordan Garcia, Mònica Balaguer Gargallo, Martí Pons-Òdena

**Affiliations:** 1Pediatric Intensive Care Unit Service, Hospital Sant Joan de Déu, University of Barcelona, 08950 Barcelona, Spain; mireia.mor@sjd.es (M.M.C.); carmina.guitart@sjd.es (C.G.P.); elena.fresan@sjd.es (E.F.R.); daniel.penela@sjd.es (D.P.S.); franciscojose.cambra@sjd.es (F.J.C.L.); yolanda.jordan@sjd.es (I.J.G.); 2Immune and Respiratory Dysfunction Research Group, Institut de Recerca Sant Joan de Déu, Esplugues de Llobregat, 08950 Barcelona, Spain; 3Pediatric Infectious Diseases Research Group, Institut de Recerca Sant Joan de Déu, CIBERESP, Esplugues de Llobregat, 08950 Barcelona, Spain

**Keywords:** lung ultrasound, bedside ultrasound, recruitment maneuvers, acute respiratory distress syndrome, pediatric intensive care

## Abstract

The use of recruitment maneuvers (RMs) is suggested to improve severe oxygenation failure in patients with acute respiratory distress syndrome (ARDS). Lung ultrasound (LUS) is a non-invasive, safe, and easily repeatable tool. It could be used to monitor the lung recruitment process in real-time. This paper aims to evaluate bedside LUS for assessing PEEP-induced pulmonary reaeration during RMs in pediatric patients. A case of a child with severe ARDS due to *Haemophilus influenzae* infection is presented. Due to his poor clinical, laboratory, and radiological evolution, he was placed on venovenous extracorporeal membrane oxygenation (ECMO). Despite all measures, severe pulmonary collapse prevented proper improvement. Thus, RMs were indicated, and bedside LUS was successfully used for monitoring and assessing lung recruitment. The initial lung evaluation before the maneuver showed a tissue pattern characterized by a severe loss of lung aeration with dynamic air bronchograms and multiple coalescent B-lines. While raising a PEEP of 30 mmH_2_O, LUS showed the presence of A-lines, which was considered a predictor of reaeration in response to the recruitment maneuver. The LUS pattern could be used to assess modifications in the lung aeration, evaluate the effectiveness of RMs, and prevent lung overdistension.

## 1. Introduction

Acute respiratory distress syndrome (ARDS) [1,2] is an acute respiratory injury characterized by important hypoxemia due to excessive alveolocapillary permeability. The pathogenesis involves pulmonary edema, diffuse cellular destruction, alveolar collapse, and disordered repair [3].

ARDS definition [4] is focused on adult lung injury and has limitations when applied to children. The Pediatric Acute Lung Injury Consensus Conference Group (PALICC) developed pediatric-specific definitions for ARDS and recommendations regarding treatment [5]. Although representing a relatively small percentage of the total number of pediatric intensive care unit (PICU) admissions, ARDS is often considered one of the most challenging patient populations for a clinician to manage [6]. 

The treatment strategies recommended in ARDS involve a lung-protective ventilation strategy using low tidal volumes and high positive end-expiratory pressure (PEEP), prone positioning, recruitment maneuvers (RMs), high-frequency oscillatory ventilation (HFOV), inhaled nitric oxide (INO), accurate fluid and hemodynamic management, and extracorporeal life support (ECMO) for refractory clinical situations [2,4].

Recruitment refers to the dynamic process of reopening unstable airless alveoli through an intentional transient increase in transpulmonary pressure [7]. Recruitment maneuvers (RMs) are ventilatory strategies consisting of sustained inflation of the lungs to higher airway pressures. This ventilatory strategy aims to reduce lung heterogeneity, improving lung collapse by slow incremental and decremental PEEP steps to improve severe oxygenation failure [6].

The safety of RMs has been proven with a low percentage of pneumothorax. Nevertheless, its impact on mortality has not been confirmed in adults [8,9,10,11]. 

Lung ultrasound (LUS) has recently emerged as an easily repeatable bedside imaging tool [12,13]. Nowadays, LUS is available in almost every intensive care unit and is a useful method for the evaluation of lung pathologies by detecting different patterns [14,15,16]. The sonographic findings indicative of ARDS include anterior subpleural consolidations, pleural line abnormalities (irregular thickened fragmented pleural line), and a nonhomogeneous distribution of B-lines [13]. 

As some recent studies have demonstrated [17,18,19,20], LUS is a useful bedside method for assessing lung RMs in patients affected by ARDS. 

This paper aims to report a case of a child with severe ARDS due to *Haemophilus influenzae* infection in which bedside LUS was used for assessing lung recruitment. 

## 2. Case

A case of a 13-month-old child who presented at the emergency department with fever and respiratory failure is reported. He was previously healthy, with no co-morbidities and optimal vaccination status. The patient presented tachycardia (169 bpm), low blood pressure (74/32 mmHg), peripheral vasoconstriction, tachypnea (64 rpm), desaturation (initial hemoglobin oxygen saturation was 56%, rising to 80% with a fraction of inspired oxygen (FiO_2_) of 100%, and fever of 39.5 °C. Physical examination revealed bilateral hypoventilation with generalized inspiratory crackles and wheezes on thoracic auscultation and profound respiratory distress. No cardiac murmur was detected, and no other abnormalities were noted.

The patient received inhaled bronchodilator by nebulization and intravenous methylprednisolone. Fluid resuscitation and vasopressors (norepinephrine 0.1 mcg/kg/min) were given for hypotension. A bedside transthoracic echocardiogram showed normal left ventricular systolic function with an ejection fraction of 60% (Figure 1). He was empirically started on broad-spectrum antibiotics. Respiratory, blood, and urine cultures were collected. The blood test showed significant leukocytosis with white blood cells of 25,790/µL (64% neutrophils, 28% lymphocytes), significant thrombocytosis with 486,000/µL platelets, and hemoglobin of 8.7 g/dL. Other laboratory features included high C-reactive protein (46.9 mg/L) and procalcitonin (43.64 ng/mL) levels. The blood gases showed a mixed acidosis (pH 7.16, pCO_2_ 60 mmHg, bicarbonate 21 mEq/L, EB −7 mmol/L) with hyperlactatemia (lactate 4.9 mmol/L). Intubation and mechanical ventilation were initiated immediately. The chest radiography revealed bilateral alveolar infiltrates (Figure 2).

After this initial management, the patient was transferred to the pediatric intensive care unit (PICU). Bronchoalveolar lavage was performed, and *Haemophilus influenzae* was isolated. No other bacterial, viral, or fungal etiologies were identified.

The diagnosis of severe ARDS secondary to *Haemophilus influenzae* infection was confirmed, and standard therapy to improve oxygenation (including a lung-protective ventilator strategy, RMs, prone positioning, high-frequency oscillatory ventilation (HFOV), and inhaled nitric oxide (iNO)) was initiated. Progressively worsening despite all adopted measures, 24 h later, he was successfully placed on venovenous ECMO.

On the 6th day on ECMO, a severe pulmonary collapse (mainly in the left lung) led to a worsening of his condition. Thus, RMs were indicated. Bedside LUS was used to monitor the maneuvers. 

The ultrasound examination was performed according to the international recommendations [21]. A 12 Mhz linear probe was used to systemically scan 6 areas for each hemithorax (superior and inferior of each anterior, lateral, and posterior zones). The findings evaluated were A-lines and B-lines, consolidations and atelectasis, the presence of a bronchogram and its characteristics, and the presence of pleural effusion [16]. LUS examination was quantified using the ultrasound reaeration score described by Bouhemad et al. [22], in which four aeration patterns were defined—N (normal aeration): the presence of lung sliding with A-lines (horizontal repetitive artifacts originating from the pleural line) or fewer than two isolated B-lines (vertical and hyperechoic comet-tail artifacts originating from the pleural line); B1 (moderate loss of lung aeration): multiple well-defined B-lines; B2 (severe loss of lung aeration): multiple coalescent B-lines; and C (lung consolidation): the presence of a tissue pattern characterized by dynamic air bronchograms.

Lung mechanics parameters were recorded: tidal volume (VT), respiratory rate (RR), positive end-expiratory pressure (PEEP), peak inspiratory pressure (PIP), FiO_2_ from the ventilator, and FiO_2_ from ECMO support.

The patient was ventilated using a lung-protective ventilator strategy in volume-controlled ventilation, chosen following the established standards [23], with the following lung mechanics parameters: TV 4 mL/kg, RR 12 bpm, PEEP 10 cmH_2_O, FiO_2_ 40%, and PIP 17–18 cmH_2_O, receiving FiO_2_ 50% from ECMO support.

At the baseline moment, LUS examination (Figure 3 and Figure 4) showed a severe loss of aeration in the left lung with dynamic air bronchograms, an irregular thickened pleural line, and multiple coalescent B-lines (C pattern). The right lung had a better aeration pattern, with the presence of lung sliding with A-lines and fewer than two isolated B-lines per rib interspace (B1 pattern).

The RMs were done using pressure-controlled ventilation and a driving pressure of 15 cmH_2_O. VT and hemodynamic status were evaluated constantly. LUS was performed simultaneously at the end of each period. 

The RMs, based on the Hodgson strategy [11], began with an increase of PEEP up to 20 cmH_2_O for 2 min, then up to 25 cmH_2_O for 2 min, and then 30 cmH_2_O for 2 min. LUS was performed at that moment (PEEP of 30 cmH_2_O), showing the presence of multiple well-defined B-lines and some A-lines in the left lung (B1 pattern) and lung sliding horizontal A-lines and fewer than two isolated B-lines (N pattern) in the right lung (Figure 3 and Figure 4). The observation of A-lines was considered a predictor of reaeration in response to the recruitment maneuver, so lung recruitment was considered effective at that point. After that, PEEP levels were decreased by 5 cmH_2_O every 2 min until the optimal PEEP (defined as the level of PEEP with the highest compliance [7]) was reached (PEEP of 14 cmH_2_O in this case). Lung mechanics parameters during the RMs are summarized in Figure 5.

A new ultrasound examination was performed at the end of the RMs (Figure 3 and Figure 4). It showed a severe loss of lung aeration with multiple coalescent B-lines (B2 pattern) but some A-lines in the left lung, especially in the anterior and lateral zones. A normal aeration pattern was detected in the right lung.

FiO_2_ remained unchanged because we did not observe desaturation during the RMs. The hemodynamic status remained stable as well (Table 1).

During the following week, there was a significant improvement in the clinical condition of the child. He was decannulated after 13 days of venovenous ECMO and extubated six days later. The total PICU stay was 27 days, and the patient was discharged from the hospital on the 46th day.

## 3. Discussion

This case report shows that bedside LUS could be used to evaluate the effectiveness of RMs in pediatric patients. The initial lung evaluation before the maneuvers showed a tissue pattern characterized by a severe loss of lung aeration in the left lung. Once the maximum airway pressure was reached, the LUS revealed an aeration improvement with the presence of lung sliding A-lines and multiple but well-defined B-lines. Then the pressure was reduced in a stepwise manner until the ideal PEEP was achieved, based on the best dynamic compliance. Concerning the beginning, with this final PEEP, the sonography pattern changed from a tissue pattern characterized by dynamic air bronchograms to a better-aerated pattern with fewer images of dynamic air bronchograms, and some A-lines in the anterior and lateral zones. There was better aeration in the right lung, so minor changes were observed. Therefore, this case supports the idea that lung aeration improvement might be detected by corresponding changes in the LUS pattern. 

The level of the highest PEEP chosen was based on the patient’s clinical and radiographic condition. Duff et al. [24] observed that RMs with a level of PEEP between 35–40 cmH_2_O for pediatric patients with ARDS can be performed safely. However, in this case LUS monitoring assisted in stopping the RMs in a PEEP of 30 cmH_2_O due to reaeration changes observed in the LUS pattern. Ultrasound can guide and personalize an open lung strategy.

Ongoing monitoring during the procedure may increase the safety of recruitment maneuvers in ventilated patients. The observation of A-lines in previously collapsed regions could be a potential marker of lung overdistension if maneuvers are not stopped at that point. The benefit of RMs needs to be balanced between reaeration and overdistention [25].

The ARDS diagnostic criteria require chest radiography (CXR) or computed tomography scan (CT) for the determination of bilateral pulmonary infiltrates [4]. Even though CXR is a very useful tool, it does not offer a dynamic monitorization and involves radiation. Moreover, it has poor sensitivity for the detection of pulmonary infiltrates compared with other imaging modalities such as CT [26].

The gold-standard method for assessing lung recruitment is CT [27]. However, it is not practical in pediatric patients admitted to the PICU for safety and ethical reasons, considering radiation issues in patients with a long-life expectancy. Consequently, there is a need for less invasive methods applicable at the bedside for measuring lung recruitment.

Ultrasound not only helps in diagnosing lung collapse and assessing hemodynamic status but also assists in the evaluation of PEEP-induced lung recruitment and lung overdistension. Compared to other tools, ultrasound requires less sophisticated skills, allowing a faster and easier learning curve [28]. The patient described was monitored by LUS at the bedside during PEEP increase until an improvement in lung aeration was observed.

In conclusion, LUS could be useful for monitoring the dynamic lung recruitment process in real-time, especially to evaluate the safety and the effectiveness of RMs and to prevent lung overdistension. It has the advantage of being non-invasive, safe, and easily repeatable.

There are some studies about RMs guided by LUS in the adult population. However, there is scarce data regarding RMs in the pediatric population, and extrapolation from adult studies is difficult due to the differences in pulmonary compliance and in the airway size in children. The presented case aims to illustrate how RMs could be safely guided by LUS in pediatric patients. Though, further studies are needed to implement it for routine clinical practice.

## Figures and Tables

**Figure 1 children-09-00789-f001:**
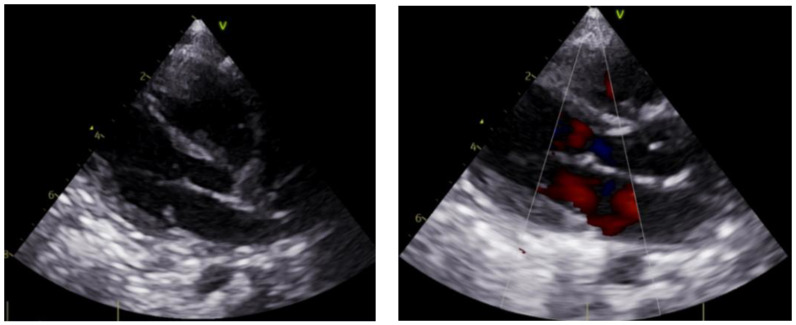
Bedside transthoracic echocardiogram showing normal left ventricular systolic function.

**Figure 2 children-09-00789-f002:**
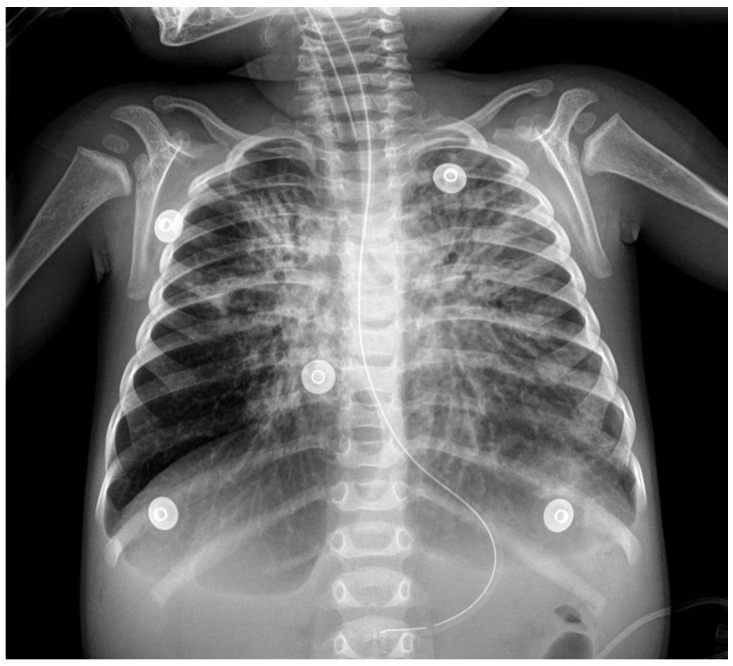
Initial chest radiography with bilateral alveolar infiltrates and tissular pattern in left lung suggesting consolidation.

**Figure 3 children-09-00789-f003:**
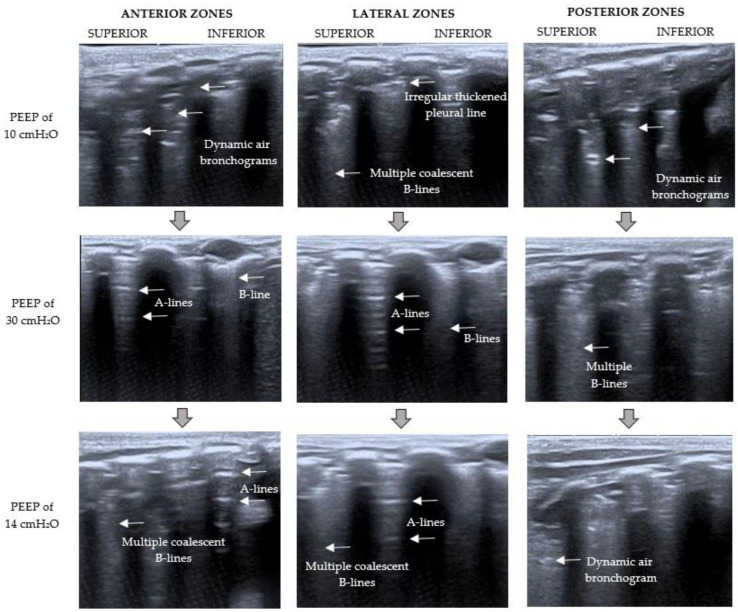
Left LUS examination during RMs. *The point where the LUS probe is applied is indicated at the top of the figure; on the left, the PEEP level at that moment is specified.* Baseline LUS (PEEP of 10 cmH_2_O) showed a C pattern (severe loss of aeration with dynamic air bronchograms). With a PEEP of 30 cmH_2_O, the pattern changed from C to B1 (moderate loss of lung aeration with multiple well-defined B-lines and some A-lines). After RMs (PEEP of 14 cmH_2_O), LUS showed a severe loss of lung aeration with multiple coalescent B-lines but some A-lines in the anterior and lateral zones (B2 pattern).

**Figure 4 children-09-00789-f004:**
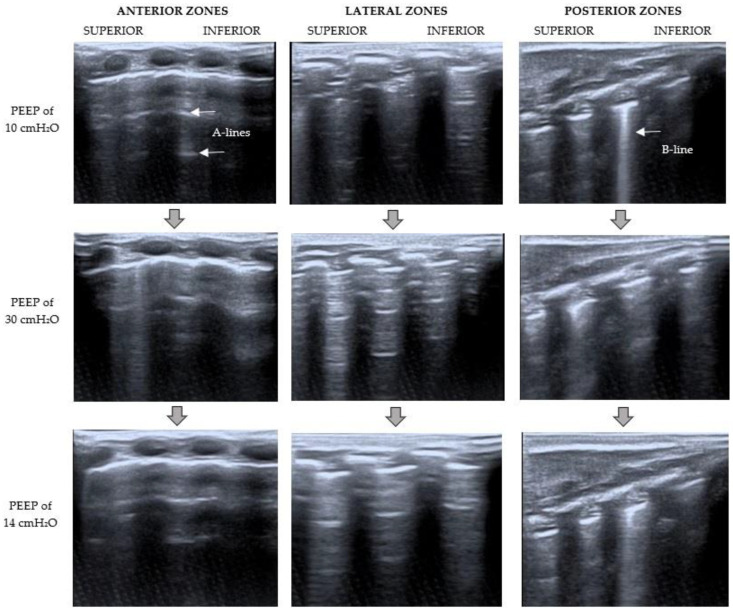
Right LUS examination during RMs. *The point where the LUS probe is applied is indicated at the top of the figure; on the left, the PEEP level at that moment is specified.* Baseline LUS (PEEP of 10 cmH_2_O) showed a B1 pattern (presence of lung sliding with A-lines and fewer than two isolated B-lines per rib interspace). With a PEEP of 30 cmH_2_O, the pattern changed from B1 to N (lung sliding horizontal A-lines and fewer than two isolated B-lines). After RMs (PEEP of 14 cmH_2_O), a normal aeration pattern (N) was observed.

**Figure 5 children-09-00789-f005:**
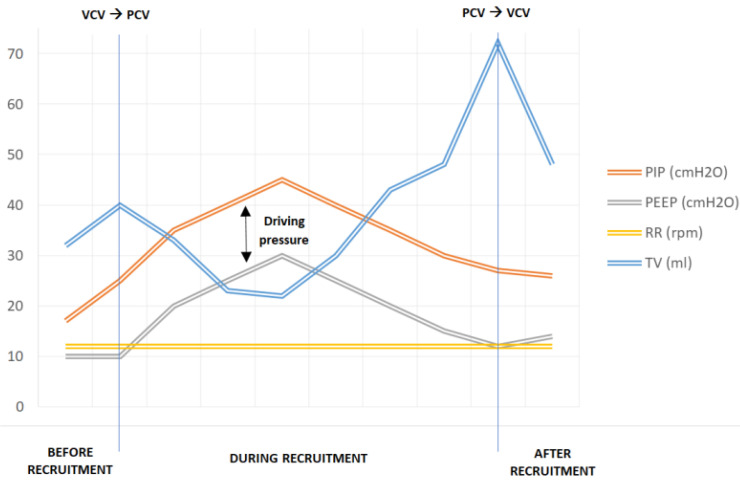
Lung mechanics parameters during RMs. PIP: peak inspiratory pressure; PEEP: positive end-expiratory pressure; RR: respiratory rate; TV: tidal volume; VCV: volume-controlled ventilation; PCV: pressure-controlled ventilation.

**Table 1 children-09-00789-t001:** Respiratory and hemodynamic status monitoring during RMs. PEEP: positive end-expiratory pressure; RMs: recruitment maneuvers; FiO_2_: fraction of inspired oxygen; ECMO: extracorporeal membrane oxygenation; HbSat: hemoglobin oxygen; HR: heart rate; MBP: mean blood pressure.

PEEP (cmH_2_O)	20	25	30	25	20	15
Time of RMs (min)	0	2	4	6	8	10
FiO_2_ (%)	40	40	40	40	40	40
FiO_2_ ECMO (%)	50	50	50	50	50	50
HbSat (%)	98	96	96	95	95	96
HR (bpm)	130	133	137	140	138	135
MBP (mmHg)	53	50	48	48	51	52

## Data Availability

The data supporting the findings of this study are available from the corresponding author on reasonable request.

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
