# Peer review of "Lung Recruitment Maneuvers Assessment by Bedside Lung Ultrasound in Pediatric Acute Respiratory Distress Syndrome"

_children, 2022, doi:10.3390/children9060789_

Round 1

Reviewer 1 Report

I am honored to review this case report from the Barcelona Group. This is a very interesting case report using bedside lung ultrasounds (LUS) for assessing PEEP-induced pulmonary reaeration during recruitment maneuvers (RMs).  RMs were indicated and bedside LUS was successfully used for monitoring and assessing lung recruitment.  The authors found that the lung ultrasound pattern could be used to assess modifications in the lung aeration, evaluate the effectiveness of RMs, and prevent lung overdistension.  Their report might have important information for all pediatrician treating infants and children with severe lung diseases such as ARDS. However, I have some major concerns about this manuscript.

Major comments

  • There have been many reports of lung evaluation by ultrasound so far. My biggest concern is the lack of difference between those previous reports and this case report. The authors should state the novelty of this case report clearly. 
  • In this case report, the authors used ultrasound to evaluate RMs for a 13-month-old child. The authors stated that the lungs initially showed a B-line on ultrasound, and the lungs showed an A-line after RMs. Did the authors have any special techniques or tips for evaluation in the very small lungs of an infant? I think it could be important for many pediatricians if this kind of information were clearly stated by the authors.
  • The LUS findings in Figure 3 are very confusing and not organized. The authors may prefer to use a simple schema that clearly shows the points where the LUS probe is applied. In addition, the LUS findings in Figure 3 should be shown with arrows which is clearly indicate A-line or B-line for the many readers who are not familiar with ultrasound to better understand.
  • One of the week points of ultrasound is that the findings are very subjective. Why did the authors not use the scoring system shown in previous reports to keep the objectivity of the ultrasound findings? This information may be important and could provide more objective information for readers.
  • Because no clear LUS evaluation criteria have been shown in this report, I couldn't identify the differences between the figures before and after recruitment in Figure 3. I agree with the idea that there is no doubt that ultrasound is useful for the evaluation of lung conditions. However, to make this method more standard, its evaluation criteria need to be clearer and more objective.

Minor comments

  • FiO2, H2O: “2” should be subscript as in FiO2 and H2

Author Response

Dear reviewer, thank you very much for allowing us to improve the manuscript. We hope we have answered all the questions. Please, let us know if further changes are required.

Please, see the attachment. We have attached three files: a cover letter to explain our responses to the reviewer, the revised version using the "track changes" function, and the new version of the manuscript.

Reviewer 2 Report

Thank you for the opportunity to review your report.  

This is a well written case presentation highlighting the use of lung ultrasound to assist in assessing the lung recruitment maneuvers in a pediatric patient. Most of the similar reports or small studies are done in adult population. 

Author Response

Dear editor and reviewer, thank you very much for allowing us to improve the manuscript. We hope we have answered all the questions. Please, let us know if further changes are required.

Point 1: This is a well written case presentation highlighting the use of lung ultrasound to assist in assessing the lung recruitment maneuvers in a pediatric patient. Most of the similar reports or small studies are done in adult population. 

Response 1: Thank you very much for your comments. Also following the comments from reviewer 1, we have added the following paragraph in the discussion: “There are some studies about RMs guided by LUS in the adult population. However, there is scarce data regarding RMs in the pediatric population, and extrapolation from adult studies is difficult due to the differences in pulmonary compliance and in the airway size in children. The case presented case aims to illustrate how RMs could be safely guided by LUS in pediatric patients. Though, further studies are needed to implement it for routine clinical practice.”

Please, let us know if this would work for you.

Reviewer 3 Report

Well designed and exposed study. Good introduction and references for a case report. Good US images and figures.

I think it should be made clear that the authors know that PEEP pressures tested are high for a 13-month-old child, but that the study was done safely because the patient was well monitored by cardio-respiratory parameters in an intensive environment.

In the conclusions, the authors mention that more studies may be needed to evaluate the usefulness of pulmonary ultrasound in titration of ventilation, but this should be more emphasized as this is only a case report.

Author Response

Dear editor and reviewer, thank you very much for allowing us to improve the manuscript. We hope we have answered all the questions. Please, let us know if further changes are required.

Point 1: I think it should be made clear that the authors know that PEEP pressures tested are high for a 13-month-old child, but that the study was done safely because the patient was well monitored by cardio-respiratory parameters in an intensive environment.

Response 1: Thank you very much for your suggestion. Few studies examine the safety and efficacy of recruitment maneuvers in pediatric intensive care unit patients. In a recent study by Duff et al, they observed that RMs with a level of PEEP between 35-40cmH2O for pediatric patients with ARDS could be performed safely. However, this is a high PEEP and, in our case, LUS monitoring assisted to stop the RMs in a PEEP of 30 cmH2O due to reaeration changes observed in the lung ultrasound pattern. Moreover, as shown in table 1, patient monitoring was evaluated constantly. We have included the following statement in the discussion: “The level of the highest PEEP chosen was based on the patient's clinical and radiographic condition. Duff et al. [24] observed that RMs with a level of PEEP between 35-40 cmH2O for pediatric patients with ARDS can be performed safely. However, in this case LUS monitoring assisted to stop the RMs in a PEEP of 30 cmH2O due to reaeration changes observed in the LUS pattern. Ultrasound can guide and personalize an open lung strategy.

Please, let us know if you would like us to explain it more.

Point 2: In the conclusions, the authors mention that more studies may be needed to evaluate the usefulness of pulmonary ultrasound in titration of ventilation, but this should be more emphasized as this is only a case report.

Response 2: Thank you very much for your comment.

With this case report, we want to share our experience related to the benefit of ultrasound monitoring in recruitment maneuvres. We think it might contribute to improve medical knowledge. Even though, more studies would be needed in the pediatric population to expand this practice. As you suggested, and also following the comments from reviewer 1, we have added the following paragraph in the discussion to emphasize it: “There are some studies about RMs guided by LUS in the adult population. However, there is scarce data regarding RMs in the pediatric population, and extrapolation from adult studies is difficult due to the differences in pulmonary compliance and in the airway size in children. The case presented case aims to illustrate how RMs could be safely guided by LUS in pediatric patients. Though, further studies are needed to implement it for routine clinical practice.”

Please, let us know if this would work for you.

Round 2

Reviewer 1 Report

I have carefully read the revised manuscript and the authors’ comments. The authors have appropriately responded to my comments and also indicated their response in the revised manuscript. I am very grateful for their understanding and for making these changes.  I think most changes are appropriate. They should consider changing one other important item. The authors stated that the aims of this case report were to illustrate how recruitment maneuvers (RMs) could be safely guided by lung ultrasound in pediatric patients. If they focus mainly on pediatric patients, this information should be included in the title as well as in the abstract.

Author Response

Dear reviewer, thank you very much for revising the manuscript again.
Thank you for your suggestion, we have included the information about pediatric patients in the title, in the abstract, and in the discussion, as you suggested.
Title: Lung recruitment maneuvers assessment by bedside lung ultrasound in pediatric acute respiratory distress syndrome
Abstract: This paper aims to evaluate bedside LUS for assessing PEEP-induced pulmonary reaeration during RMs in pediatric patients.
Discussion: This case report shows that bedside LUS could be used to evaluate the effectiveness of RMs in pediatric patients.
Please, see the attachment.
Thank you for your work and dedication.
